# Heritage Sites, Devotion, and Quality Enhancement in Tourism: The Promotion and Management of Ancient Marian Places of Worship along the Appian Way in Puglia and Basilicata

Luigi Oliva [1,*] and Anna Trono [2]

1   Ministero della Cultura, Parco Archeologico dell'Appia Antica, 00185 Roma, Italy
2   Department of Heritage, University of Salento, 73100 Lecce, Italy; anna.trono@unisalento.it
*   Correspondence: luigi.oliva@cultura.gov.it

**Abstract:** Religious tourism is a significant and growing field of tourism that overlaps with cultural tourism. It has the potential to improve the quality of life of those who live in places of faith or along routes of spiritual interest. Religious tourism involves a complex interplay of spiritual and economic motivations. Effective religious tourism management requires respect for spiritual values, partnerships, local engagement, and quality assessment. Devotional practices have evolved from medieval spiritual care to communal expressions and periodic rituals. This paper specifically analysed the characteristics of the Marian cult and pilgrimage flows to places of Marian faith. It examined their value potential from a religious and cultural perspective and their role as a particular attractor of experiential and quality tourism generated by the territorial context. The area of reference is the region of Puglia, which has often played the role of cultural bridge with the eastern coasts of the Mediterranean in the past. The second part of the paper focuses on the proposed itinerary along the Appian Way in its final route between Puglia and Basilicata. Marian shrines were sometimes the cause and sometimes the evidence of the cultural and economic poles that characterised the medieval and modern variants of this ancient road route. The study outlines a serial path that integrates the usual settlement or infrastructural levels of territorial knowledge with the Marian theme, which was analysed diachronically. An operational track in the contemporary territorial dimension emerged from the correlation of both the stratigraphic reading of the landscape and the interpretation of material and immaterial cultural heritage. This track aims to aggregate and promote the sustainable rediscovery of those places, which are largely cut off from the routes of mass tourism, in adherence to the most recent European and local cultural and landscape guidelines.

**Keywords:** spiritual routes; contemporary pilgrimage; Marian routes; Marian shrines; local worship of virgin; Puglia religious itineraries; Basilicata religious itineraries; Appian Way

## 1. Introduction

The reconstruction of a sanctuary geography linked to the cult of the Virgin is a research topic that has gained momentum in recent decades at both national and regional levels (Rech 2017). A recent study by the École Française de Rome in collaboration with thirty Italian universities has catalogued Italian sanctuaries (http://www.santuaricristiani.iccd.beniculturali.it/AreaPubblica.htm, accessed on 8 December 2023). In 2012, the international research project RECULTIVATUR, which aims to develop a tool for including religious-related cultural values in the planning and development processes of urban centres, systems of settlements, and surrounding rural areas, launched the cataloguing of Apulian sanctuaries (https://recultivatur.gtk.uni-pannon.hu/ accessed on 8 December 2023). This research topic connects current traditions, the identity dimension, and community practices that still strongly characterise the region to a past of interregional relations, cultural models, land organisation and management, pilgrimages, and transhumance (Sensi 2003; Calò Mariani 2003, 2019a).

As part of the range of goods and services on offer, places of worship are also a means of communication and can be understood as a cultural element of regional characterisation and attraction. They enhance the location's visibility and renown, improving and supporting its reputation; they create, adapt, consolidate, and publicise the region's image; they increase social cohesion and the local community's sense of belonging; and they maintain traditions and activate the growth of economic and social components.

This is also true of religious sites in general, which increasingly represent the destination of an educated and quality tourism arising from the growing demand for experiential and educational travel but also from the broadening of the range of regional goods and services on offer, which are centred on uniqueness and a strong local identity.

Travel towards places of worship continues to be of great relevance. To a large extent it coincides with cultural tourism, but it is able to beget other new forms of tourism. Those who practise it are in search of places of faith linked to culture and local cultural heritage. They are attracted by the landscapes in which the religious sites lie, often linked to religious festivities and special events of a religious nature. They seek to visit structures of religious significance and historical and/or artistic importance, such as cathedrals and monasteries, famous sanctuaries, churches, and abbeys. However, there is also growing interest in small churches and little-known places of worship, which often constitute the fabric connecting sites of greater attractiveness, forming a network with links of varying importance. An example is the devotional pilgrimage to the church of Santa Maria di Pierno, a centre of worship of Our Lady of the Assumption, which has maintained its vitality in the surrounding area, especially San Fele. The pilgrimage takes place in an area spanning the boundary between Puglia and Basilicata, delimited to the east by the plateau of the Murge hills and to the west by the Bradanic depression. Another example is the sanctuary of Santa Maria di Banzi, a salient feature of which is the wooden statue of the Virgin, patron saint of Genzano di Lucania, which sees the arrival of pilgrims each year linked to the celebrations of her feast day. Also highly popular is the Sanctuary of the Madonna di Picciano, situated on a relief in the foothills of the Murge, in a highly evocative landscape.

Such places of faith highlight not just a region's religious, cultural, and landscape value, but also act as attractors of experiential and other forms of quality tourism generated by the regional environmental context. They thus represent a "complex cultural item", which can make a substantial contribution to regional development. This entails identifying and harmonising the synergistic effects that derive from the close complementarity of cultural and tourism services, aimed at both consumers, and associated commercial activities. Cultural and religious tourism thus represents an opportunity for local development, making it possible to diversify the "reasons for visiting" and to link the holy places to other items of interest, both cultural (food-and-wine, handicrafts, folklore, architecture, etc.) and natural.

Geographical, historical, and material analyses of selected significant places of Marian worship in Puglia and Basilicata have shed light on specifically local elements and cultural circulation that find a semantic correspondence in the religious dimension and a deep-seated regional continuity in the roads that link the sites. The study and interpretation of these epiphanies constitute the starting point for a new eco-compatible religious and spiritual tourism that improves the quality of life of those who live in places of faith or along routes of spiritual or religious interest, projecting an image of them that is attentive to the principles of environmental sustainability.

## 2. Fulfilling the Potential of Places of Worship and Routes of Faith

The last few years have seen a reorganisation of the range of regional tourism goods and services on offer that focuses on new forms of tourism generated by the growing demand for experiential and educational travel. These new (and not so new) tourism models include places of faith and cultural and religious routes and itineraries, which are the historical and contemporary expression of a complex regional heritage, rich in religious,

cultural, and landscape values and meanings. The multiple actors, activities, resources, and skills involved in them activate a systemic process for extracting the maximum value from each item that extends to the entire regional range of goods and services. The religious heritage item becomes an aggregating factor of cultural and economic interest, able to generate new entrepreneurial opportunities via the development of complementary products and services (Rogerson 2007). This enables an equitable distribution of the income from tourism among various local economic operators (Meyer 2004) and favours the development of the local economy as a whole (Mariotti 2011; Beltramo 2013).

Making the most of places of faith and religious routes in economic terms raises the issue of their management, drawing attention to the organisation and unity of a system of actors, which in the tourism field entails the interaction of subjects pursuing divergent interests and objectives. Appropriate governance in this sphere is not easily achieved because it requires not just regional knowledge but the ability to go beyond the logic of competition. It is necessary to stipulate agreements between the various actors, both religious and secular, public and private, entrepreneurs and ordinary consumers, local and national, in order to create a shared vision and encourage the participation of all those subjects in the destination's overall design. An important role is played here by local stakeholders who consent to join in a network and create a system of quality tourism. Their involvement makes it possible to maintain social cohesion in destinations and to try out new communication strategies between interested parties. The participation of stakeholders, or rather the "heritage community" (Faro Convention), is important because it is based on a strategic communication perspective, which confers decision-making power on them and favours social dialogue based on the principles of participation, solidarity, and responsibility. "Responsibility" in this context is linked to the empowerment of local populations, who, by means of a learning process, achieve self-determination, authority, and co-participation in the management of their heritage. We are thus dealing with the conscious bilateral use of heritage, concerning both supply and demand, both hosts and guests, the content of which is exquisitely cultural. The result is a circuit that on the supply side entails going beyond the sector's economic limits and looking at the social content. This starts with the cultural identity of the host regions and peoples, whose awareness is raised thanks to a tourism demand that is itself more aware, in a virtuous circle of awareness, satisfaction, and well-being on the part of both sides of the hospitality equation. On the visitors' side, there is the recovery of the value of slowness, as well as learning about the context and themselves. "Such a circle is the necessary and sufficient precondition for the creation of a positive feedback relationship between supply and demand, a condition and a sign of a resilient and self-reinforcing regional tourism system: sustainable and capable of lasting and evolving positively over time" (Tinacci Mossello 2014).

The particular attention paid to the religious item is not always coterminous or in harmony with the ethics of those who "appropriate" it, especially if they are inclined to interpret religious motivation above all from an economic point of view, considering the weight of the religious factor in the market. Indeed, it sometimes happens that pilgrimages towards holy places and spiritual tourism do not live up to the motives and socio-cultural processes that originally gave rise to them and supposedly sustain them today. This clash of principles involves a tension between tradition and modernity, showing how the spiritual motivation that guides those processes can involve more than a simple ritual and is in fact bound up with the interests and interactions of diverse categories of subject. The communion between cultural expressions and religious sentiment, in other words the search for both spirituality and contact with art, nature, and beautiful landscapes, should be reconcilable with the sustainable development of sacred places, avoiding conflict with the needs arising from commercial logic. This then tends to create new market niches, new types of supply and demand, new circuits, new entrepreneurial players, and new wealth, ultimately constituting a dynamo of economic development in the places being visited.

It is indeed true that needs of a religious and economic nature are not always compatible, and the dynamics governing the relationship between worship and culture are

not always simple and straightforward. Valerio Pennasso (Beltrami 2017), head of the National Office for Ecclesiastical Cultural Heritage and Buildings of Worship of the Italian Bishops' Conference, points out that the perceived evolution of churches into museums (with or without the need to buy a ticket) is partly the result of social and economic shifts unfolding in the secular world, and above all, an approach to cultural heritage that is generated by mass tourism, which devours holy places as it does all consumer products. This legitimises the recourse to charging for entry, which serves to regulate the flow of tourists, while reserving parts of the church for silence and prayer. The rules of the Canonical Code and the measures applied by the ecclesiastical authorities clarify when the practice might be adopted, mainly seeking to limit and discourage it (Feliciani 2010; Azzimonti 2016), considering the religious motive to be paramount. However, in the opinion of many religious authorities, there is a conflict between religious needs and those pertaining to the management, conservation, and restoration of cultural heritage, operations that are typically financed from the public purse using national and international resources, giving rise to much disquiet.

Demand for visits to sacred places is increasing, as is the range of goods and services offered by a vast and varied range of operators: a universe of private companies in tune with the latest developments in the international tourism market, characterised by exceptional competitiveness and capacity for forming strategic alliances with the main actors in the sector, both public and private (Di Maio and De Simone 2006). The focus on the environmental impacts is fuelling debate over the sustainability of religious travel, especially the business aspect of those who organise religious journeys and provide services in holy places, who are prone to trivialise the religious motive and subordinate it to purely economic interests (Shackley 2001). The problem is that the "discovery" of religious tourism, together with the flow of tourists and public funding that frequently promote and support it, raises expectations not only in the religious world but also in the secular sphere. The latter is composed not just of travel agencies and cultural associations, but also of actors who are skilled in managing the religious aspect in accordance with pragmatic and utilitarian criteria, exploiting the sacred locations and images in a poor-taste blend of religion and marketing, conferring on the holy place a business rather than a spiritual role.

The choice of promotional strategies is therefore highly important. They must aim to respect the religious motives that generate religious travel in the first place; incentivise forms of partnership between the secular and religious spheres; strengthen participation and the capacity for acting on a local level; guarantee the quality and efficiency of the goods and services on offer; and, last but not least, analyse and assess the cultural and economic results in quantitative and qualitative terms (Trono 2020).

### 3. Marian Places of Worship: Some Case Studies in Puglia and Basilicata Regions

*3.1. Religious Tourism, Places of Faith, and Marian Routes*

Marian routes represent an interesting field of experimentation for the development of religious itineraries that can be used to test the interplay of spiritual and economic motivations. Unlike many saint cults, which are linked to specific periods or cultures, the Marian cult has shown a substantial continuity of appeal from the Middle Ages to the present day, in some contexts even continuing previous pagan cults.

Among the numerous devotional routes are those that adorn the image of the Virgin Mary with all aspects of female excellence and rich symbolic content. The Marian devotion of the community is centred on the story of the bond of love between Mother and Son. It is a story of the redemption of oppressed humanity, of salvation in the face of natural disasters. It is the story of justice on this earth. It is an authentic, collective story of lives lived, knowing no distinctions of class, gender, or age: entire communities identify with their Madonna (Ruppi and Nuzzo 2008). Interest in Mary is extremely ancient. It is seen as early as the 4th century in Palestine and Alexandria, in the buildings dedicated to Her worship. Her tomb was the site of early pilgrimages to the Holy (Marella 2014), as can be

seen in the wall paintings of the catacombs and in the prayer Sub tuum praesidium written on an ancient Egyptian papyrus dated to the late 3rd century AD.

Marian devotion is founded not only on the homage paid to Her exceptional dignity and saintliness but also on the unceasing and confident appeal to Her power of intercession with Jesus Christ. In Her, the Church recognises divine maternity and the role that Christ has assigned to Her in the economy of salvation. The Madonna is the Mother of God, Theotokos, and the Mother of Christ Christotokos. «The humanity to which the Blessed Virgin Mary gave birth was always that of the Son of God himself. That is the reason why the Assyrian Church of the East is praying [to] the Virgin Mary as "the Mother of Christ our God and Saviour". In the light of this same faith the Catholic tradition addresses the Virgin Mary as "the Mother of God" and also as "the Mother of Christ"». Mary thus becomes a sign of unity: She is Christotokos but also Theotokos. The man Christ of which the Assyrian Church of the East speaks is God and Saviour, given human form by Mary for salvation, the divine son, which is a Trinitarian stage that unfolds in the Church, «body of Christ and temple of the Spirit», via the sacramental process (Bruni 2009, p. 271). Following the Council of Ephesus of 431 AD, She acquired the title of Dei Genetrix or Theotokos (Mother of God) and subsequent Councils conferred on Her the role of Mother. With the recognition of the dogmas of Her divine maternity and virginity, Mary has always occupied an eminent place in popular worship and piety (Maritano 2006). The evolution of Marian worship, along with the liturgical forms and the progressive reawakening of Marian sanctuaries, is strongly linked to the establishment in 1854 of the dogma of the Immaculate Conception, which followed centuries of division and conflict. More recently, it has been driven by various factors, some political (arising from tensions between politics and religion in the early 20th century) and others social and cultural.

Europe has the densest network of Marian sanctuaries, the target of short pilgrimages undertaken by thousands of worshippers (Marian Studies 2000; Moved by Mary 2009; Bugslag 2019).

In the Apulian-Lucanian regions, the links with the East are evident. Mediated by the Byzantine world and the dominant cultures of Frankish-Germanic origin, the latter influenced by the evolution of the Roman Mariology, devotion to the Virgin Mary was consolidated above all in rural contexts, where it often played a key role in rites linked to the fertility of the earth and human beings, the rhythms of the seasons and agricultural activities, protection from danger, and Her intercession in the forgiveness of sins.

Marian worship is practised throughout Puglia and Basilicata. Numerous images of the Virgin, frequently depicted using Her healing powers, are found in sanctuaries situated in evocative natural contexts, along the roads trodden by the pilgrims (including the southern stretch of the Via Francigena, subregional itineraries such as the roads leading towards Santa Maria di Leuca, and interregional itineraries such as the network of migratory herding routes) or near the ports from which they embarked for the Holy Land, including Siponto, Barletta, Bari, Brindisi, and Otranto (Calò Mariani and Pepe 2013; Calò Mariani 2019a, 2019b).

### 3.2. Defining a Geographical and Methodological Path of Research

This study investigated the cultural heritage in a territory of southern Italy that is divided in two administrative regions, Puglia and Basilicata. Due to its administrative division, the area is often studied in isolation, leading to a fragmented understanding of cultural phaenomena that can only be fully understood from an interregional or even transnational perspective. For example, some cultural trends in the region may have originated elsewhere and been adapted to local conditions, while others may have been exported to other regions. By studying the region's cultural heritage in a broader context, we can gain a better understanding of its significance and its role in shaping the wider cultural landscape of southern Italy. The church of Santa Maria di Anglona is a prime example of how recent studies have helped to redefine our understanding of its artistic and constructional history (Pace 1996).

Episodic or focal analysis is a methodology that aims to define territorial references that can integrate the usual settlement or infrastructure levels of knowledge of the territory with thematic and periodic levels. The systematisation of these levels offers new keys to reading the stratigraphy of the landscape and interpreting the traces of the past fossilised in the present.

The methodology of research combines historical research with critical analysis of structures. The episodes chosen for analysis are dedicated to specific manifestations or epithets of the Virgin Mary (Immaculate, Assumption, Nativity, and Justice). These are connected to the seasonal circularity and rhythms typical of the agricultural-pastoral reality, or to the relationship between faith, ethics, and justice, which was fundamental to supporting the social order before the Enlightenment. The case studies are representative of different development processes of Marian sites: some of them are still relevant in the regional context, some only local, some other lost all the religious interest being defunctionalised or abandoned.

Other criteria used to select episodes include their extra-urban dimension and ability to attract faithful and pilgrims from distant areas; the complexity of their architectural and functional stratification, testifying to their vitality in different epochs and continuity of use; and their representativeness in relation to management (religious orders, military orders, commanders, dioceses).

To provide a practical experience in the field, the itinerary develops in daily stages along the medieval variants of the Appian Way from the deep hinterland of Lucania to the coast of Taranto. This testifies to the change in the landscape within a system that was once much more culturally and economically connected than it is perceived today.

### 3.2.1. Geomorphological and Geological Characteristics of the Study Area

The physical landscape of the area plays a pivotal role not only in shaping the development of settlements and infrastructure but also in influencing the prevalence and resilience of religious practices. These practices are inextricably intertwined with the region's socioeconomic conditions, which are themselves profoundly shaped by the terrain, its resources, and the challenges of survival in sometimes isolated and challenging environments.

The region stands on the Murge plateau to the east and the Bradanic depression a tectonic basin to the west. The study area is characterised by its development between two rocky formations that define an unstable and fragmented alluvial valley floor. The succession of emergences and sinkings in remote geological epochs has accumulated significant layers of tuff, clay, and sand, alternating with large areas characterised by thin soil dotted with white limestone rocks. These latter are found at the origin of the appellative "petrosa" (stony) that accompanies many place names in the area.

The calcareous nature of this area is responsible for the scarcity of significant watercourses, which are absorbed into the lower layers in the form of true underground rivers. This phenomenon, known as karst, has produced a particular territorial morphology that has always conditioned the entire settlement dimension. Shallow furrows incise the slopes and can extend onto the shelves, locally known as "lame". These furrows constitute a dense network of almost always dry torrent beds, often cultivated in their bottom. The rocky walls, from very remote times, have hosted hermitages or troglodyte dwellings, with a density that reaches its peak near the Gulf of Taranto with deep ravines, locally called "*gravine*" (Zezza and Zezza 1999). Nestled within the rugged depths of the gravine, a captivating chapter of religious devotion unfolded during the Middle Ages. Carved into the embrace of these natural amphitheatres, a tapestry of cave settlements emerged, each a testament to the region's deep-rooted faith. Within these sacred spaces, a unique religious devotion blossomed, intricately woven around the veneration of saints and peculiar Marian epithets.

### 3.2.2. Historical and Human Influences That Shaped the Layout of Settlements

Starting from the Middle Ages, the crossing of the region, in periods when the necessary political and economic stability was guaranteed, was strongly influenced by seasonal

climatic variations or by the particular characteristics of the journey, such as religious, official, or military ones (Dalena 2022), or those linked to the transhumance of sheep (Bavusi and L'Erario 2016).

The great impetus of the medieval Agricultural Revolution gradually ran out as a result of the tax burden imposed by Frederick II and the first Angevin kings, the endless feudal wars, the demographic collapse corresponding to the general negative situation that involved the whole of Europe in the 14th and 15th centuries. In the southern context, the crisis must be attributed to the progressive conversion of the model of articulation of the countryside towards an organised system of farms for the extensive exploitation of the territory under the pressure of an increasingly globalised international market. In fact, starting from the late Norman period, the rise of rich communal economies in the north of Italy, based on advances in proto-industrial production techniques and a monopoly on commercial traffic, led to a widening gap between the north and south of the country. This gap was only partially contained by the development of primary production in the south, despite timely protectionist interventions and incentives towards the development of indigenous entrepreneurial classes by the various dynasties in power (Tramontana 2000).

Among the consequences of the crisis was the destruction of the network of "casali" (farm villages), the real engines of the medieval economy: in large numbers, they were abandoned under the growing pressure of urbanisation, supported by royal policy (Tourbert 1997, pp. 301–15).

Starting from the late Modern Age, even the phenomenon of livestock transhumance underwent a gradual halt, and with it a strong element of cultural continuity that on several occasions overlapped with the phenomenon of religious travel, inheriting many material aspects and devotional practices. As the ultimate consequence of these factors, the system of churches and chapels scattered in the countryside for the "*cura animarum*" (literally: care of souls) began an irreversible decline, and with them their strong ability to catalyse and root the rural population, increasingly attracted to the large settlements (Licinio 1983).

### 3.2.3. Religious and Anthropological Context

Marian devotion is often linked to several founding legends, almost all characterised by the finding and transfer of images of the Madonna or by miraculous events associated with them (Verrastro 2007; De Palma 2010). Funding legends have a strong local attraction power and, in an ideal devotional geography, also transregional. Some studies notice that those ancient legends seem to support the hypothesis that sanctuaries played a role in controlling territory and social pacification (Verrastro 1999). Nevertheless, the cult of the Virgin has been associated with a relationship with God that escapes the conditioning of power, a kind of aspiration of the faithful to a space of freedom and a less controlled religion (Vauchez 1993, p. 472). This explains the intrinsic strength of the Marian cult, the spread of its practice, and the predominance and prevalence of it over other saints' cults.

### 3.3. Case Studies

The following subsections are the results of a wider research that studies some local or regional attracting Marian shrines placed on the path of the Appian way or along its medieval variants (Trono and Oliva 2021) (Figure 1).

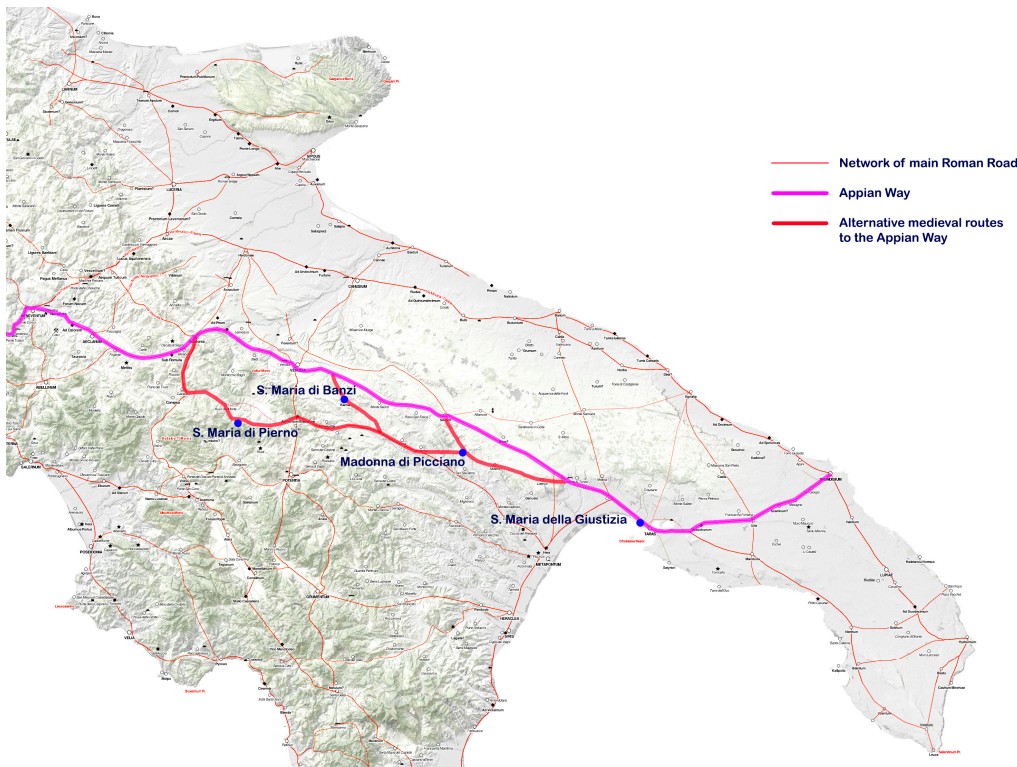

**Figure 1.** Localisation of case studies along the Appian Way and the alternative medieval routes on a map of major Roman roads in the area.

### 3.3.1. S. Maria di Pierno

S. Maria di Pierno (40°47′33.2″ N 15°36′17″ E) is a unique and significant example of a medieval monastic complex located at the crossroads of important trade and pilgrimage routes. Its well-preserved architecture and rich history offer a valuable window into the religious and cultural life of southern Italy in the Middle Ages. It stands on a plateau at 960 m above sea level, located about ten kilometres from the town of S. Fele, at the foot of the looming rocky mass of Mount Pierno (Figure 2).

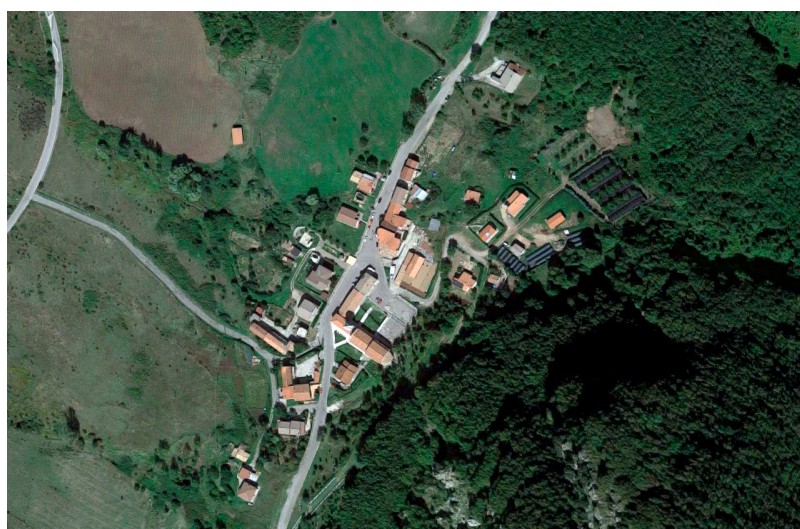

**Figure 2.** Monastic complex and sanctuary of Santa Maria di Pierno (at the centre). Aerial view of the settlement (Source: Google Maps 2023).

Nowadays, it is still the annual destination of an impressive pilgrimage to the sanctuary of the Virgin (Pascale 1970). The site is hidden by a natural crown of dense forests that preserve the appearance of the original Lucanian inland landscape.

Apparently remote from the main internal Lucanian ways, Pierno is located at the crossroads of two important routes: the first allowed direct connections between the thriving centres of the Vulture (Melfi, Venosa, Rapolla, Atella, Acerenza) and the Roman via Popilia (Forum Popili) or its coastal Campanian variant (Paestum); the second, at the edge of the Pierno heights, connected the centers along the Appia (Guardia dei Lombardi, Lacedonia) with Potenza, joining the Roman via Herculia for Grumento and the Ionian coast.

According to the founding legend, Guglielmo da Vercelli built the church near the place where he found an image of the Virgin (Verrastro 1988), finding partial confirmation in the news of the donation of the church (therefore, presumably already in existence) to the Saint by the bishop of Rapolla in 1132. The period corresponds to the phenomenon of eremitism and the itinerancy of religious people that interested the southern regions, starting from the 11th century, in the Italo-Greek cultural context that characterised them (Caputo 1996).

In a privilege of 1141 of the diocese of Rapolla, Bishop Ruggiero I describes the church as «obedientie monasterii Sancti Salvatoris de Goleto» (Fortunato 1968, p. 34), the important female monastery founded by Guglielmo da Vercelli near S. Angelo dei Lombardi. The grancia (the term refers originally to a fortified granary in the country but later was extended to large farmhouses ruled by monasteries) was made up of "oblati" directed by a prior appointed by Goleto, within the protection and favour of the Balvano, lords of Vitalba and Armaterra. That political and territorial context reveals the pivotal role of the monastery in relation to the fiefdoms of the feudal lords and the development of internal road connectivity in Norman times. The land that the church still owns in the rural village of Pierno and the surrounding countryside are the remnants of its ancient territory (Giustiniani 1816, p. 163).

Among the benefactors of the monastery was Riccardo di Balvano, justizier and contestable of the kingdom, who in 1175 and 1187 (Pedio 1964, p. 12) donated possessions and goods. Inscriptions on the gate of the church report that his son Gilberto II commissioned the renovation of the church between 1189 and 1197 to the master stonemasons of Muro Lucano under the direction of the Sarolo brothers (Fortunato 1968, p. 20). Margherita, widow of Gilberto II and last member of the family, donated, between 1198 and 1200, a substantial part of the fiefdom of Vitalba. In the meantime, Goleto, with all its dependencies, was welcomed by Pope Lucius III in «apostolicam protectionem», freeing it from obedience to the local bishops (Kehr 1962, p. 515, n. 113).

In the late Middle Ages, S. Maria di Pierno increased its influence, acquiring ever greater autonomy from the mother house. In 1514, by the will of Leo X, the convent was promoted to a monastery placed under the patronage of the Caracciolo, first, and the De Leyva, later (1552), to whom the latter owe important works of renovation and decoration (Cianciulli and Pepe 1781). In 1614, a pastoral visit led by the diocese of Melfi shows the near-complete abandonment of the complex by the verginiani (Murana seu Melphiensis Iurisdictionis 1712, f. 167–69), while a 1673 description commissioned by the new commendatari, the Doria, reveals the structural and functional integrity of the church and annexes. Another description from 1711, linked to a dispute between the dioceses of Muro and Melfi, highlights the presence of accommodation facilities for travellers, confirming the sanctuary's continued use as a pilgrimage site, despite the closure of the convent (Murana seu Melphiensis Iurisdictionis 1712, f. 206 v). That late description provides evidence of the typical functional layout of monasteries. In particular, the guesthouse, located near the main entrance to the oratory, was separated from the monastic complex and belonged to an area reserved for relations with the outside world. A topographic map of 1793 shows the state of the premises on the eve of the 19th- and 20th-century transformations (Angelini 1988, pp. 24–26).

Two excavation campaigns, conducted in 1997 and 1998 by the Operational Office of the Archaeological Superintendency of Muro Lucano in the square in front of the sanctuary brought to light two superimposed levels, adapted to the slope of the site. The archaeological data show that the stratigraphy can be divided into five main phases:

1. Original foundation, not dated with certainty.
2. Phase of wealth and expansion, from the Swabian period to the end of the Middle Ages.
3. Renovations of the Modern Age (from the 16th century).
4. Degradation and downsizing of the structures, from the 18th century.
5. Restorations after the earthquake of 1980 (Cappiello and Pagliuca 1999).

Today, substantial parts of the ancient complex are still visible, including the liturgical part, consisting of the bell tower, porch, and church, which was partially reorganised in the 16th century (Figure 3a); the monastic part, with some buildings; and the reception area, with the annexes and fountain.

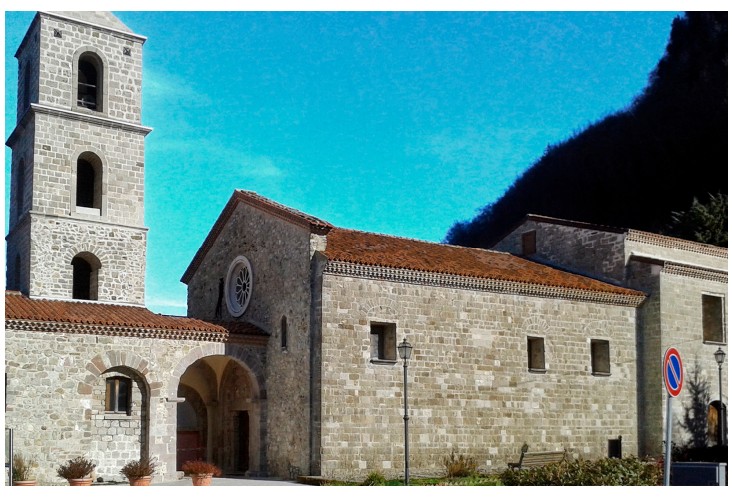
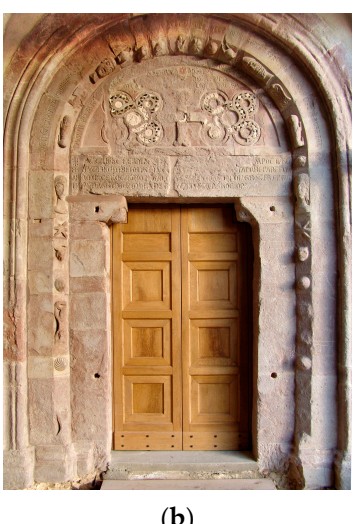

(**a**)  (**b**)

**Figure 3.** Monastic complex and sanctuary of Santa Maria di Pierno. (**a**) Detail of the southern façade with the entrance arch. (**b**) Main entrance portal to the church.

The porch with the church's portal, the mature fruit of the assimilation of Romanesque plasticity and stylistic elements in the Apulian-Lucanian area with elements of peculiarity and innovation, can certainly be attributed to the intervention of the "*magistro*" Sarolo between 1189 and 1197. In the lunette, the adoption of mosaic spirals with a strong Byzantine reference is cleverly combined with the naturalistic high reliefs that frame the arch, infusing it with depth and emotion (Figure 3b) (Manchia 2017, pp. 83–89). The strong figurative association with coeval sculpture in the Gargano area of Puglia (especially those in the Marian churches of Calena and Pulsano) can be also related to the mobility of workmen and artisans along the main pilgrim route of the Appian Way that connected Rome to the shrines of St. Michael and St. Nicholas and the Adriatic Sea ports.

The church is equally peculiar for its Modern Age extension. In fact, a rear hall was added to the original three-naved body with four spans, placed in spatial continuity with the first following the breaking of the apse choir. The columns present elements of recovery, elements of early medieval tradition, and parts attributable to the master craftsmen of Muro, especially in the bases. The recent restoration has restored the original wooden roof, set with a longitudinal structure supported by transverse arches discharging on corbels. Restorations and excavations have highlighted the two-level structure of the complex, highlighting the exploitation of natural slopes to create a series of communicating semi-subterranean rooms, still partly visible and used by private individuals who reside in the

surviving buildings of the monastery or pilgrims. Other archaeological evidence regards the production of bells in the Angevine period (Peduto and Lo Pilato 2007).

Despite the long period of closure to the public, made necessary by the damage to the structures caused by the earthquake of 1980, the cult of the Assumption of Pierno, whose feast falls on August 15, has retained all its vitality in the surrounding territories and in San Fele. With the reopening of the complex in 2001, the resumption of the frequency of the sanctuary was matched, with the support of the Diocese of Melfi–Rapolla–Venosa and the Holy See itself. In 2004, Pope John Paul II crowned the simulacrum of the Madonna in the Vatican during the 150th anniversary of the proclamation of the Dogma of the Immaculate Conception (Spera 2003).

At the beginning of the year 2023, religious organisations, local governments, and associations signed an agreement regarding the creation of the Walk of San Guglielmo da Vercelli (1085–1142). The Walk follows the travels of the Catholic hermit and the founder of the Congregation of Montevergine through the Marian sanctuaries of Montevergine, Goleto, and S. Maria di Pierno to the port of Barletta. The agreement aims to have the trail included in the "Catalogue of Italian Religious Walks" of the Italian Ministry of Tourism. (https://piccolipaesi.com/category/il-cammino-di-guglielmo/, accessed on 8 December 2023).

### 3.3.2. S. Maria of Banzi

The monastery of Banzi (40°51′42.3″ N 16°00′50.4″ E) stands on the ruins of ancient Bantia, known from sources as a Roman municipality from the 2nd century BC (Figure 4). The medieval geographer Guidone, in tracing the main routes of the region, includes it along a variant of the Via Appia, on the road that linked Acerenza to Muro Lucano (Guidone 1990, p. 124). The ager bantinus played an important role in the articulation of the territory in Roman times: the link with the Via Appia has recently been investigated in a more systematic way, also in relation to the ministerial and UNESCO programs for the enhancement of the so-called *Regina viarum* (Mutino 2022).

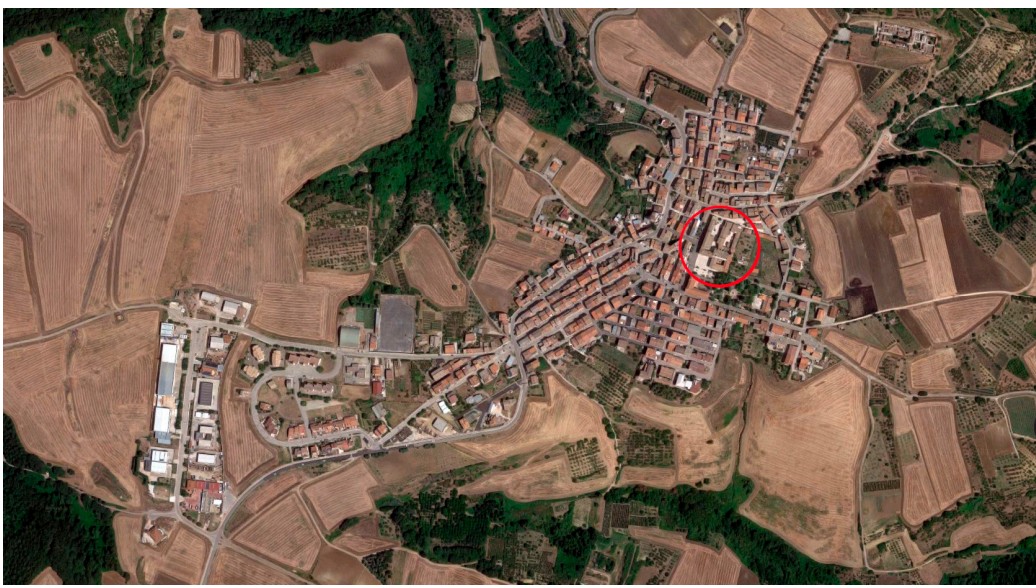

**Figure 4.** Aerial view of Banzi. The monastic complex and sanctuary of Santa Maria di Banzi is highlighted at the center of the residential area (Source: Google Maps 2023).

The Memorie del monastero bantino is a manuscript written in 1755 by Domenico Pannelli, secretary to the commendatory abbot of the Abbey of S. Maria of Banzi. It traces a detailed description at the time and transmits precious documents that allow

us to reconstruct the main phases of transformation and management of the complex (Pannelli 1995).

The first known document dates back to the year 798: it is the donation of the "monasterium Sancte Marie in Banze situm in finibus Acerentie cum servis et ancillis atque colonis" by Grimoaldo III, prince of Benevento, to the abbot of Montecassino (Gattola 1734, p. 19). The question of the real dependence on Montecassino remains somewhat uncertain. A controversial document of 1081 reports the donation, made by Goffredo count of Lecce, of the church of S. Maria di Banzi and its dependencies, to the Benedictine monastery of Cava (Guerrieri 1895, p. 64). In 1088, at the request of the abbot, Pope Urban II placed Banzi under the direct dependencies of the Holy See, freeing it from any subjection to the diocesan bishops and allowing only the burden of a tax due annually to the archbishop of Acerenza from the church of S. Anastasia that the abbot of Banzi owns in Acerenza (Pannelli 1995, p. 68). In 1089, Pope Urban II consecrated the church of Saint Mary, under the urging of the Norman princes and Abbot Ursone. On that occasion, a Holy Door was opened in the city walls, on the axis that led to the church (Codice Diplomatico Barese, I 1897, p. 61, no. 33).

Between the 11th and 12th centuries, the abbey became a very important center within the Norman fiefdom, acquiring possessions in Puglia as well. From the mid-14th century, the first signs of decline were felt, culminating in the cession of the abbey to a commendatory abbot in the 15th century. From that moment on, the monastery lost its territorial power, even if the church retained its sanctuary value for the Marian cult. In 1536, it passed to the Augustinian Order, and in 1688 to the Reformed Friars Minor of San Francis of the Province of Basilicata, who, with the favour of Cardinal Barberini, began the expansion and renovation of the church to the south (Pannelli 1995, p. 96; Borraro 1978, pp. 27–31). The construction of the new monastery took many decades (Marotta 1972). After the suppression in 1866, and the inclusion in the municipality of Genzano di Lucania in 1904, the structures underwent numerous transformations to adapt them to new uses (De Bonis 1999, pp. 125–30).

The church underwent considerable changes, both due to the poor wall consistency-made of opus incertum, with thin layers of mortar and materials, mostly salvaged, with irregular pieces, and due to the degradation to which the roof materials were subjected.

During the urgent restoration work made necessary by the effects of the 1980 earthquake, some evidence of the ancient church was brought to light, including two fragments of the precious early medieval mosaic floor, depicting a lamb with the symbols of the four evangelists and ornamental motifs. Some frescoes from the Benedictine factory and part of the capitals of the original cloister also emerged (Civita 1992, pp. 1007–16).

Excavations have revealed that the original layout consisted of a three-nave plan divided by six square pillars, with two semi-pillars attached to the wall of the three-apsed presbytery and two in the counter-façade; the floor was decorated with mosaics, as evidenced by the finds of substantial mosaic fragments in limestone and terracotta, with mostly zoomorphic representations (Salvatore 1996).

The current church layout reveals the spatiality linked to the renewed liturgical sensibility, particularly Franciscan, with a single nave and no external apses. Despite the creation of a late Baroque decorative apparatus, the structures reflect the ideals of simplicity and economy of the Reformed Franciscans. Some elements dating back to the medieval period are also still visible, including the 12th-century porch, with doors and two mullioned windows, two 15th-century windows, and parts of the cloister.

The Banzi settlement's distinctive Marian vocation, already evident in the abbey's dedication, is underscored by a series of precious artworks preserved in the parish church. These include the statue of the angel, possibly Gabriel, and the 14th-century bas-relief embedded in the façade (Figure 5a), depicting the enthroned Virgin and the patron saint (Figure 5b); the 13th-century Marian icon on panel; the wooden statue of the Virgin; and the renowned triptych (Madonna and Child, St. Peter and St. John the Baptist) by Andrea Sabatini da Salerno, dated 1517 (Noviello 2014, pp. 294, 367). The Madonna serves as the

town's patron saint, and on 8 September, popular devotion still draws many faithful to the annual patronal festivities.

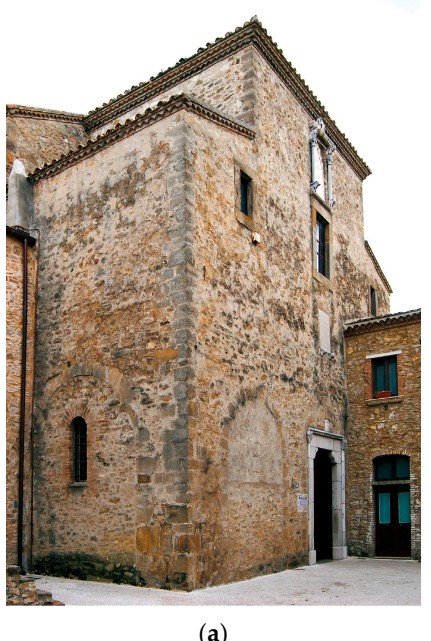 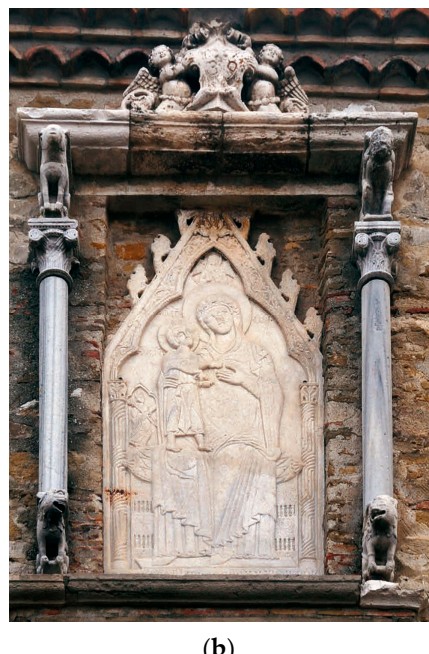

(**a**)  (**b**)

**Figure 5.** Monastic complex and sanctuary of Santa Maria di Banzi. (**a**) Main façade of the church. (**b**) Detail of the bas-relief depicting the Virgin.

### 3.3.3. Sanctuary of Madonna di Picciano

Similar to the previous ones, the Sanctuary of Madonna di Picciano (40°41′56.3″ N 16°28′21.1″ E) stands on a hill at the intersection of ancient ridge routes that formed the current road system (Figure 6). The path that runs alongside the site is in fact one of the medieval alternative route (called diverticula in the late Latin sources) of the Appian Way, connecting Banzi, Gravina, and Matera, which became essential during the Middle Ages (Guidone 1990, p. 124; Dalena 2003, p. 57) when Matera acquired nodal importance for both travel and the transhumance of livestock.

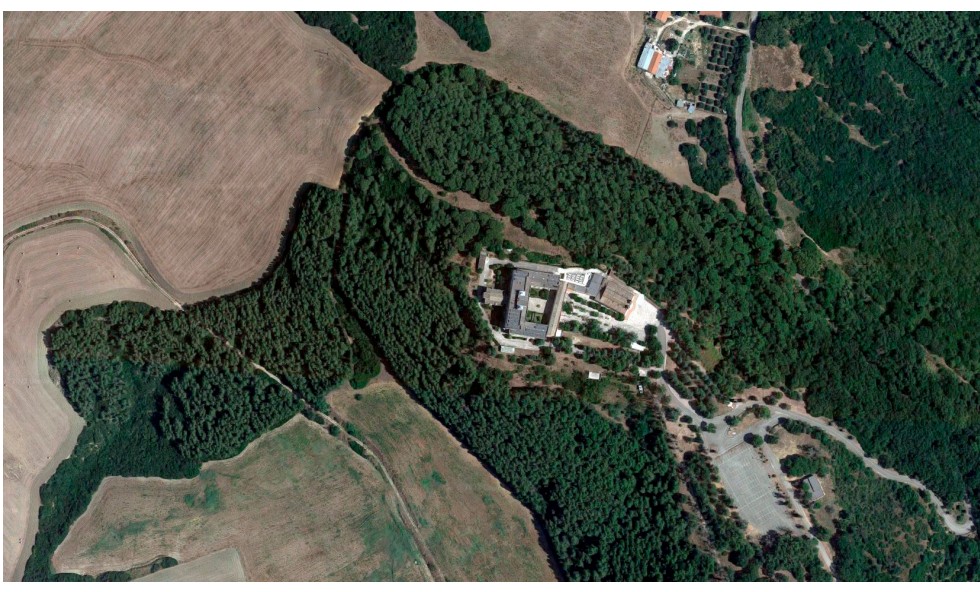

**Figure 6.** Sanctuary of Madonna di Picciano (at the center). Aerial view (Source: Google Maps 2023).

Near the shrine, numerous natural or artificial caves can be found that are attributable to the widespread phenomenon of rupestral culture, which has been attested, in various ways and different periods, from prehistory to the present day. Among these, the crypt of Grottini is noteworthy, but there is no evidence or sources to attest to a link with the subsequent subdial monastic settlement (Padula et al. 1995).

The first documents in which the monastery appears date back to the 13th century. They are contracts and bulls involving the abbots, dated 1219, 1238, 1252, 1254–1258, and 1273. These documents reveal a significant level of importance already at a local level, a sign of the fortunes of the Benedictine monastery, in which the son of King Tancredi, William III of Altavilla, the last Norman ruler, was buried. He died presumably before 1199 (Panarelli 2010, p. 67). In 1279, the church is mentioned as the property of the Knights Templar (Garufi 1933, p. 40, n. 31), although recent research has shown that the Benedictines remained in the complex at least until the early decades of the 14th century (Copeti 1982, p. 269; Montesano 2019). On 12 March 1308, the Templars Dominic Turrosa, Angelo da Brindisi, and Stefano di Antiochia were captured "*in domo Piczani*" and transferred to Barletta (Prutz 1888, p. 363). In 1332, the monastery appears as a grangia belonging to the Gerosolimitana domus of Giovinazzo (Vendola 1939, p. 999). "*Frater Ludovicus*" gerosolimitano appears in the late 14th century as "*Praeceptor Picciani*" (Tansi 1746, p. 96). Under the Hospitallers, the settlement was adapted to the characteristics of fortified commendas, with a system of defensive posts and a building for the residence of the Commendator (Giordano 2011).

Under the commendator Girolamo Carafa, important work was carried out, including the reversal of the orientation of the church and the transfer of the apse fresco with the image of the venerated Virgin. The transfer highlights the devotional importance of the simulacrum. Under Silvio Zurla (1642–1685), the main altar was rebuilt with the tabernacle in which the fragment of fresco was placed, and new side altars were built (Archive of the Monastery of Picciano, Cabreo della Commenda S. Maria di Picciano, n. 6024, year 1674, f. 9v). Finally, Pierantonio Gaetani was responsible for the addition of the right-side aisle, completed in 1794 (Volpe 1818, p. 223).

The Commendator had civil and criminal jurisdiction in the territory of Picciano; he was exempt from local duties and had the right to appoint four chaplains under his direct control. Therefore, Picciano, like the other Hospitaller commendas, can be configured as an administrative island belonging to the archipelago of the assets of the military orders, distributed throughout Christian Western Europe. The presence of monastic military orders has often been associated with pilgrimage phaenomena, both as religious-sanctuary and as service and hospitality infrastructure.

The Commenda of S. Maria di Picciano was suppressed from 1807 to 1816, under the bailiff fra Giuseppe Caracciolo di Santeramo (Gattini 1928, p. 31).

After the unification of Italy, the sanctuary was acquired by the municipality under the protection of a custodian and a chaplain appointed by the king. In 1956, the hospital for pilgrims was built, and in 1962, after agreements with the local municipality, a convention was signed between the Archdiocese of Matera and the Olivetan Congregation, which acquired the Sanctuary with its dependencies and undertook to guarantee the continuity of the cult of the Virgin, the assistance and care of souls (Campoli 1987; Campoli et al. 1996).

One of the oldest descriptions of the Marian devotions on the hill of Picciano is provided by Eustachio Verricelli in 1595: «il dì della Nonciata a 25 di marzo se fa la festività con molto concurso dei forastieri per lli grandissimi miracoli che fa...» (Verricelli 1987, p. 61). A cabreo dated to 1674 reports that the pilgrims who came from the surrounding area on the day of the fest were about twelve-thousand. The passage of pilgrims and faithful is also attested by the rich collection of graffiti that can be found on the walls (Camarda 2019). A recent study has shed light on the contribution of clergy and pilgrims to the development of the complex (Giordano 2015).

In the late medieval period, the growing importance of the city of Matera and the sub-region of the gravine determined a strong attraction factor, able to divert the Roman route of the Appian Way and to produce a series of strategic settlements for the control of

the territory and the traffic. Among them, Picciano, ruled by the military orders, can be considered a focal point in the territory.

Inside the complex, which was heavily renovated and modernised during the past century, the church stands out, having preserved some medieval vestiges and the subsequent interventions of the Modern Age that were carried out on behalf of the Hospitaller commanders. The building has a basilica plan, and the interior is characterised by an archaic spatiality, marked by the evidence of the masonry masses. The minimum height of the round arches on pillars that separate the naves contrasts with the rise of the massive, pointed vault of the central nave, probably a late remake of a previous vaulted structure. The need to increase the primitive settlement determined functional changes to the original layout, leading to the inversion of the orientation of the church, obtained by breaking through the apse, of which there are traces of masonry texture and fragments of fresco on the current counter façade. A recent study on the symbolism of the Romanesque portal preserved in the chapel dedicated to the processional statue of the Annunciation placed it in the apocalyptic theme of the Day of Judgment, of Norman influence (Centonze 2017). The fragments of fresco found on the apse show numerous devotional graffiti, names and dates left by faithful and pilgrims. This decoration probably included the venerated image of the Virgin, which, detached and initially placed in a side altar, is now at the centre of the dossal of the main altar. The painting, dated to the end of the 15th century and attributed to a southern painter, is of Adriatic culture.

The new façade, erected in correspondence with the previous apse and dome, was built without decorations, juxtaposing the sloping roofs of the side aisles to a central body crowned by two bell towers at the top. The effect, whether intentional or the result of an interruption in the decorative program, is one of military austerity, essential and defined by pure volumes, according to canons that are found in other buildings commissioned by the military orders in the region (Oliva 2014) (Figure 7a). On the west side, the shrine is flanked by a later construction that forms a kind of ambulatory, with a barrel vault that passes behind the current presbytery, in front of the old entrance portal. From this ambulatory, where the statue of the Madonna di Picciano is kept, it is possible to access the chapel of the Pietà, the sacristy, and another service room (Figure 7b). In the ambulatory behind the presbytery, a chapel has been created for the devotion to the statue of the Madonna di Picciano. This devotional statue, which could be dated to the early 1700s, is commonly attributed to the generosity of the shepherds, who came in transhumance in the area. Following the model of the sacred icon, the statue reproduces the Virgin Mary half-length, in an attitude of prayer, placed on a nimbus, symbol of the transcendent world in which the Mother of God was assumed. In 1953, a fire severely damaged the ancient image of the Virgin, which was replaced by the current one (Giordano 2015).

### 3.3.4. Santa Maria della Giustizia in the Surrounding of Taranto

The monumental complex consisting of the church of Santa Maria della Giustizia (40°29′23.14″ N, 17°11′27.085″ E) with the adjoining convent stands a few kilometres west of the city of Taranto, on the western side, where three ancient Roman roads branched off: the Appian Way, the Traianea jonica, and the via per compendium to Bari (Guidone 1990, p. 124; Oliva 2007) (Figure 8). It represents an example of a religious structure in which converged aspects of popular devotion, elements of hospitality and service related to its location with respect to the roads, and economic exploitation of large areas of territory. The structure stands on the edge of the rocky terrace that runs parallel to the coast, raised about 15 m above the underlying coastal strip, about four kilometres from the current estuary of the Tara River. Having lost its religious and agricultural functions, the complex is now incorporated into the industrial area, between tanks of the ENI refinery and hydrocarbon treatment plants, which, due to their widespread emissions, despite the restoration campaigns carried out and planned in the next years, make it impossible to use the building for a continuous time.

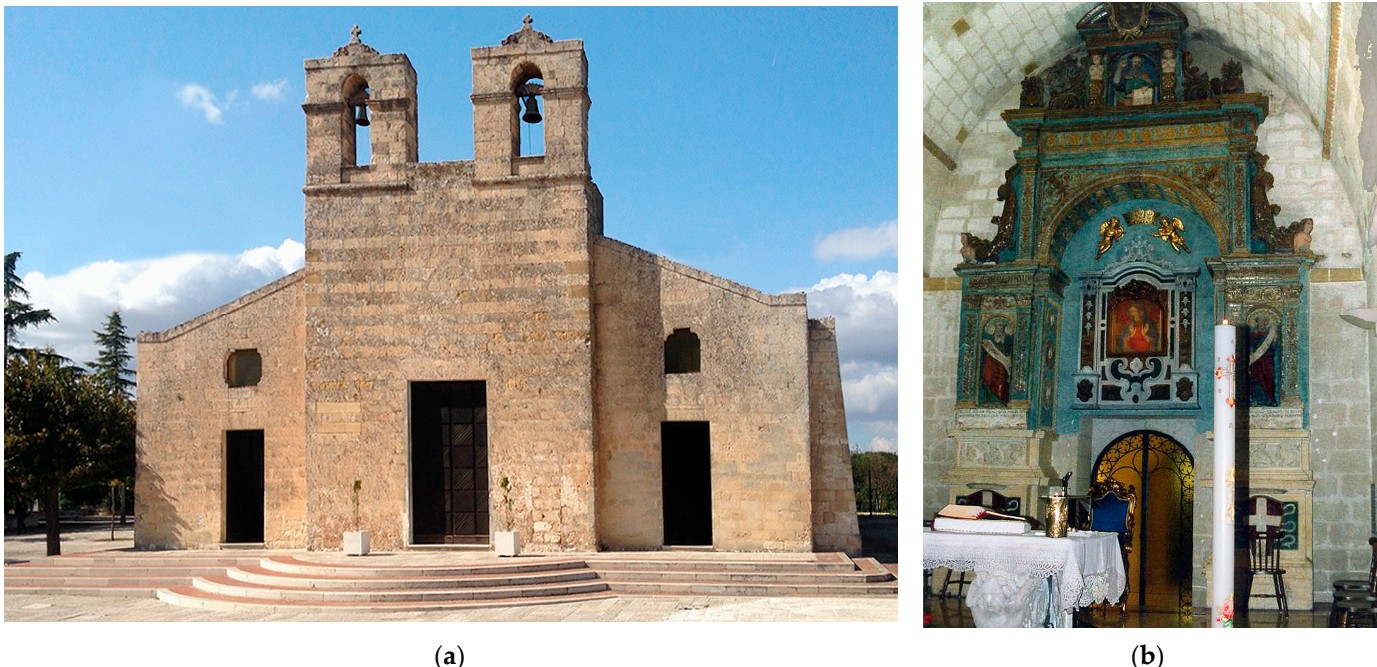

(**a**)  (**b**)

**Figure 7.** Sanctuary of Madonna di Picciano (at the center). (**a**) Main façade of the church (**b**) Detail of altar of the Virgin.

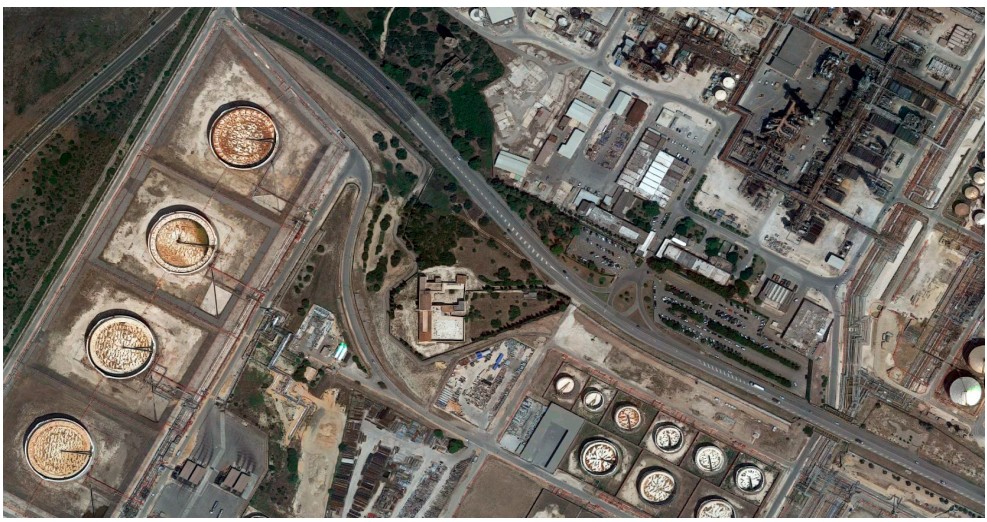

**Figure 8.** Monastic complex and sanctuary of Santa Maria della Giustizia (at the centre of the industrial area). Aereal view (Source Google Maps 2023).

Recent historiography does not deviate from the studies of Valente (1897) and Blandamura (1927, 1928a, 1928b), who report a series of documents that are mainly lost. The first of these concerns a concession dating back to 1119 in which Constance of France, widow of Bohemond I, and her son, the young Bohemond II, donated a land to the Greek monastery of S. Pietro, which arose on the larger of the two islands that make up the small archipelago of the Cheradi off the coast of the city (Von Falkenhausen 1993, pp. 133–66; Fonseca 1977, pp. 83–108) placed "*in portu nostro civitatis Tarenti iuxta basilicam Beati Nicolai de Vetraniolo, ut in praedicta terra domum praefatus abbas suique successores aedificent ad utilitatem ipsius sancti monasterii et peregrinorum pro Dei amore peregrinantium*" (Tanzi 1902, p. 134). Another document, issued by Roger II in Troia on 15 September 1131, confirms the concession of these possessions on the Tara River (Blandamura 1928a, p. 43). A subsequent

document attests to the actual realisation and activity of this hospice for pilgrims, it is a bull of Clement III which confirmed the possessions of the Greek monastery among which it mentioned the sanctuary "*cum hospitali et domibus in locum Tarae [. . .] quam a bonae memoriae nubili muliere, uxore quondam Boemundi [. . .] concessam*" on the condition that the monastic community, while continuing in the Greek rite, observed the rule of Saint Benedict (Holtzmann 1962, p. 236). Of the subsequent phases relating to the interventions of the 13th and 14th centuries, it is not possible to provide any further documents. Given the legendary dimension and the late documentation of the dedication story, the reason and date of the change in dedication of the complex proposed by Blandamura in 1194 (1928a, p. 38) need to be critically reconsidered.

A concession from the Archbishop of Taranto, Cardinal Giovanni d'Aragona, dated 4 September 1482, entrusted the church to the Order of Monte Oliveto: "*Ecclesia S. M. de Iustitia sic titulata, seu noncupata, sita extra moenia Civitatis Tarenti, sit et fuerit de Mensa Maioris Eccl. Nostre Cathedralis Eccl. Tarentine et ad ipsam semper pertinuerit et pertineat ius dominijs*" (Blandamura 1925, pp. 257–59). In the document, the site consisted only of a small church dedicated to the Virgin, which was the object of local veneration due to a miraculous event that gave rise to the epithet, which occurred during the time of Giacomo del Balzo in the late 1300s. The chapel was occupied by a ninety-year-old hermit, Cataldo Firlicius, and his non-Christian servant, Samuele. One day, the servant rebelled against the hermit and killed him, but he remained immobilised near the body of his master for three days, during which a violent downpour rained incessantly on the city. On the third day, the prince, who had arrived at the place at the invocation of the same servant, first baptised him and then executed him.

In the 16th century, the monastery was damaged by Saracen raids (Lubin 1693, p. 180) and, later, fortified and restored with the aid of royal subsidies (Blandamura 1928a, p. 50). After further looting, the Olivetans, in 1725, obtained permission to move within the city walls on condition that the church of Justice did not remain completely abandoned (Archive of the Archbishopric Curia of Taranto ACAT, Shelf VIII, I, 20, 4).

After the suppression of the Order in 1808, the site was first acquired by the State and then sold to be used as a farmhouse.

In the 1980s, the Superintendence for Environmental, Architectural, Artistic and Historical Heritage of Puglia conducted a restoration campaign that involved the demolition of all the superfetations dating back to the adaptation to the farmhouse, in order to allow the reading of the convent's architectural configuration. During the cleaning work, traces of frescoes emerged, which were restored onsite (Ressa et al. 2003, p. 11). Archaeological excavations conducted in 1995, 2015, and 2022 revealed evidence dating back to the Roman era and the Middle Ages (structures and agricultural pits, burials).

The complex is developed around two large, square courtyards. On the western side stands the church, whose longitudinal body extends to occupy a corner of the courtyard, with the two façades enriched by decorated entrance portals. The building is oriented the apse facing west and the main façade facing the regional road axis. The remaining side is delimited by the refectory, which perpendicularly adjoins the church. The relationship between this austere external façade, the development of the robust perimeter walls, and the refined decoration of the openings on the inner courtyard, although limited to some episodes at the first floor, provide an idea of the condition of bulwark in which the settlement had to develop (Figure 9a).

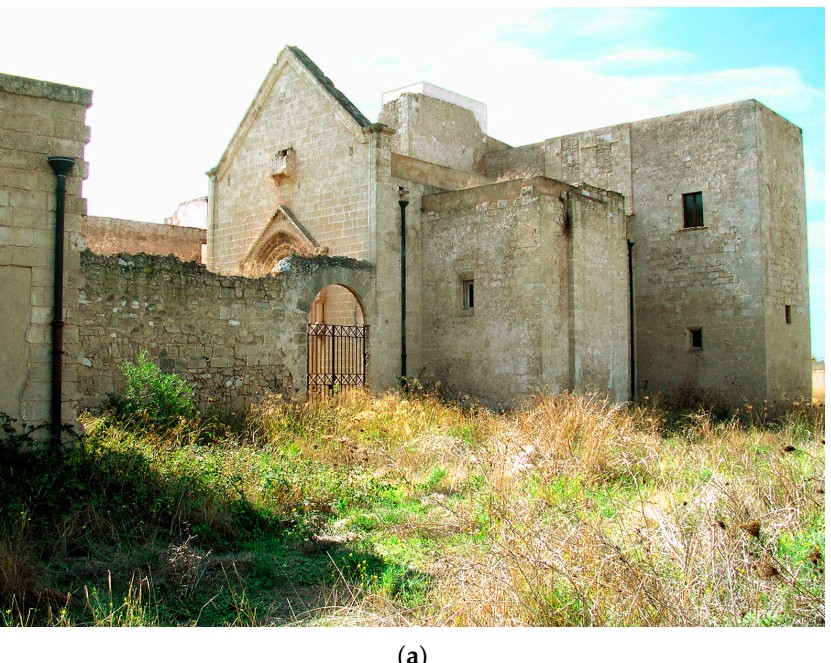
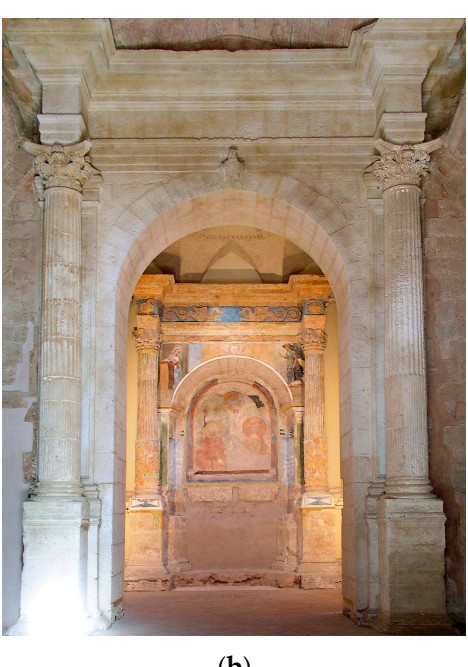

(**a**)                                                                                                 (**b**)

**Figure 9.** Monastic complex and sanctuary of Santa Maria della Giustizia. (**a**) Exterior view of the complex from north. (**b**) Detail of the monumental entrance arch to the chapel of the Virgin. The palimpsest fresco of the Madonna is visible on the back wall.

The church is a unique example in the local panorama of a full preserved original structure (Tocci 1975, pp. 201–08). The façade is articulated with a cusp framed by a perimeter torus cornice that folds back about one meter from the ground to run parallel to the ribbed pillars of the portal. This type is common in the churches of Taranto that have been erected or renovated during the Angevin period, of which the façade of S. Domenico in Taranto is the most striking example (Oliva 2021). The side portal seems to be a reduced and simplified version of the main one, from which it is distinguished by the insertion of a precious, archaic architrave decorated with stylised floral motifs. That decoration seems to recall Syrian patterns that can be found in area of Byzantine influence, probably conveyed in the region by the import of furnishings and urns (Caprara 1979, pp. 93–102).

The interior of the church is mainly preserved in its original layout, except for the 16th century chapel, covered by a beautiful star vault, dedicated to the veneration of the Virgin's image. The building type is a single nave, divided into two ribbed vaults with almond shaped ribs that recall Cistercian and military orders models.

All the structures can be dated between the mid-14th century and the 17th century.

The fresco palimpsest depicting the icon of the Virgin, placed in the only modern chapel (Figure 9b), dates back to the 12th century. It was probably relocated from the previous church, where the miraculous event took place.

The chapel of the miraculous Virgin was a destination for pilgrims until the abandonment of the monastery by the Olivetans. Despite the dramatic change in the function of the complex, its architectural and decorative components have survived, likely due to the presence of the sacred image of the Madonna della Giustizia and the complex's strong internal coherence.

## 4. Conclusions

### 4.1. Marian Sanctuaries and Historical Landscape—A Starting Point

Marian worship is widely attested in the Puglia-Lucania region during the Middle Ages, initially strongly influenced by Byzantine religiosity, then conveyed by the dominant cultures of Franco-Germanic origin conditioned by the evolution of Roman mariology (Au-

gias and Vannini 2015). It became particularly consolidated in the Modern Age, especially in the extra-urban area, where the Virgin often assumed a key role in the scansion of the rites linked to the fertility of the land and people, the regular alternation of the seasons and activities, protection from danger, intercession for the remission of sins, and the religious foundation of earthly justice.

In the context of the rich body of studies on sanctuaries and cults, Marian ones occupy a special position for their diffusion and for the cultural relevance linked to the different forms of representation and veneration of the Virgin. The episodes presented have been studied in a singular and non-comparative way, in order to avoid forced analogies between cases. Compared to thematic or travel guides, this approach has implicitly revealed the anthropological and cultural reasons underlying the birth and development of each settlement, which are strongly conditioned by natural elements and the construction of the landscape. It has also presented some of the manifestations of these reasons, which are inevitably linked to the individual and local historical component.

Starting from the premises, on the level of devotion, it is possible to formulate two considerations that open up to working hypotheses:

1.  The passage from the Middle Ages to the Modern Age and from this to the Contemporary Age has marked a change in the characteristics of devotion, from the original *cura animarum* for the medieval agricultural context, to the symbolic and community value embodied by the sanctuary poles, to the festive and processional periodicity of today.
2.  Where the cult was not related to particular groups or ancestral rituals of the place but was sponsored by more political-economic factors or was located in more dynamic and open areas, it has been greatly reduced or has disappeared altogether.

On an architectural level, the preservation of the original place of worship or at least of its medieval appearance, regardless of cost, is clearly linked to a hierophanic logic based on the development of legends or miraculous events that inextricably bind the simulacra of the Virgin to the places of veneration and to the territorial context. Following the historical facts, the architectural transformations of Marian shrines are driven by a variety of factors, including changing needs and the desire to create a more visually appealing and accessible space for worship.

Finally, it should be noted that, even in historical vicissitudes and periods of lower attendance, in the presence of an integrated and continuous community, the sacredness of the places has almost always been respected, preserving the sites from contemporary looting and speculative destruction.

*4.2. The Promotion of Tourism: Risks and Opportunities*

There are numerous places of worship dedicated to the Virgin Mary in the analysed region, whose attractiveness lies not only in the buildings but also their surrounding context. Of great artistic and architectural value, they were the subject of frequent visits in the past but today lie abandoned, often unknown to even the local inhabitants, despite having a potential in terms of local development that is almost "palpable". They are the expression of local culture and spirituality, but they are also a tourism "product" whose potential has yet to be fulfilled. Their strategic and promotional value is perhaps greatest in inland and marginal areas characterised by small shrines consecrated to popular devotion that are the object of local micro-pilgrimages.

Their recovery could give rise to a well-organised tourism system able to coordinate religious structures and re-connect the buildings and their history with the reality of their host region, whose development was in many cases shaped by them over the centuries. By identifying the synergies arising from the close inter-complementarity of religious, cultural and tourism services aimed at consumers and their associated economic activities, it will be possible to generate quality tourism that both recognises the values of the soul and facilitates regional development.

However, the presence of travellers in holy places and along the routes of faith requires careful planning and management. Respect must be shown not only for the heritage items' religious and cultural nature and the visitors' faith and culture, but also for the meaning of that heritage for both the administrators of the site and the host community, if necessary, modifying the way in which the image of the religious item is communicated, without altering its character or content. The management of religious tourism is a relatively new paradigm, and as with any new model, it is still evolving and requires the cooperation and involvement of all institutions, both religious and secular.

It is thus necessary to construct a network of alliances on multiple levels of regional government, in accordance with a logic that envisages valid planning, opportune promotion, and attentive management. Planning means drawing up a good programme in which research, the study of the region and its tourism dynamics, and the use of innovative technologies is combined with the knowledge of the consumers themselves, the commitment and capacity of every stakeholder involved, and the entire system of promotion of religious and spiritual tourism. Promotion must be suitable for situations that involve the cultural and religious experience. It must also take account of the need for rational, shared solutions that improve behaviours and the quality of life and produce useful and sustainable results for the community and the region as a whole. Management entails careful monitoring of holy places, but it also requires professionalism and respect for ethical and cultural values. Sustainable religious and spiritual tourism is desirable in the interests of the heritage itself, the tourist, and the local community.

**Author Contributions:** Methodology, L.O. and A.T.; Validation, A.T.; Formal analysis, L.O. and A.T.; Investigation, L.O. and A.T.; Resources, L.O. and A.T.; Writing—original draft, L.O. and A.T.; Writing—review & editing, L.O.; Visualization, L.O. All authors have read and agreed to the published version of the manuscript.

**Funding:** This research received no external funding.

**Institutional Review Board Statement:** Not applicable.

**Informed Consent Statement:** Not applicable.

**Data Availability Statement:** No new data were created.

**Conflicts of Interest:** The authors declare no conflict of interest.

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
