# Peer review of "Heritage Sites, Devotion, and Quality Enhancement in Tourism: The Promotion and Management of Ancient Marian Places of Worship along the Appian Way in Puglia and Basilicata"

_religions, doi:10.3390/rel14121548_

Round 1
Reviewer 1 Report
Comments and Suggestions for Authors
The article presented here is a well-documented and interestingly presented study of the geographical analysis of the Marian devotion that occurs in the territory of Puglia and Basilicata / Italy.
Both the objective and the methodology were indicated clearly and correctly. The analysis has been carried out in detail and with a geographical view of the subject under study.
The article reads well and in principle I have no major comments to make on it.
The only minor additions they suggest making are at two points:
1. I am not entirely convinced of the relevance of the subsection relating to the geology and geomorphology of the area. How does this relate to the subject of the analysis?
2. Figure 1. The text refers to the location of the study sites along the Appian Way. Perhaps it would be useful to mark the Appian Way on this cartogram?
Author Response
Dear Reviewer, first of all, I would like to express my sincere gratitude for your appreciation of our article.
With regard to your valuable suggestions, we found both to be highly beneficial in enhancing the comprehension of the analytical process and the contextual description.
- Specifically, to clarify the rationale behind the introduction of the geomorphology paragraph, we propose incorporating the text with a specification that these elements, in our opinion, constitute a pivotal factor in the development of settlements, the interpretation of cultural relationships, and the dissemination of unique and recurrent cults within that geographical region.
- With reference to the Figure 1, we have developed a new image based on a geomorphological plan, which represents the route of the Roman Appian Way in the study area and the alternative path developed in medieval times that still exist today. The relation among medieval variations, geomorphology the network of minor roads of Roman times, and the presence of sacred places is the main theme of the context analysis .
We hope that the new version of the manuscript will meet your observations.
Reviewer 2 Report
Comments and Suggestions for Authors
Focusing on the southern Italian regions of Apulia and Basilicata, the article aims to analyse the characteristics of local Marian cults and pilgrimage flows and their potential for developing and managing regional experiential and qualitative tourism.
While this topic is potentially of interest to various academic debates on pilgrimage and religious tourism, the author fails to present a clearly structured discussion and coherent argument that would contribute to current debates in this field of knowledge. In a sense, the rather long title already points to the structural problems of the paper and its lack of clear focus and rigid analysis. For example, the hypothesis is presented towards the end of the article. Also, the content from lines 642-741, or parts of it, could be flagged earlier on to provide a clearer idea about the aims and objectives of the research.
The article draws on some of the existing literature on religious tourism but leaves its focus on Italian literature unexplained. Indeed, there is very little scope for discussion and critical readings of already existing studies and debates outside of Italy. This reviewer would have appreciated more information on this seemingly narrow focus.
The article shows considerable conceptual weakness and lacks a clear methodology. Religious and cultural tourism both as conceptual constructs as well as fields of study are also not fully explored within the context of wider debates. The author may wish to engage here more specifically with landscape and heritage tourism rather than the more malleable concepts of cultural and religious tourism.
The author appears not to address a wider and cross-disciplinary readership but rather a more specialised audience. The author should place his/her analysis in the context of a particular debate and/or academic subfield. This would help to sharpen the analytical focus of the article and make it more accessible to a wider readership.
Comments on the Quality of English LanguageEditing of English language required.
Author Response
Dear Reviewer,
first of all, we would like to express our sincere gratitude for your suggestions regarding our manuscript. We appreciate the time and effort you have dedicated to thoroughly reviewing our work. Your detailed comments and constructive criticism have been instrumental in guiding us towards a more comprehensive and well-structured presentation of our research.
In particular (citation of Comments and Suggestions for Authors in quotes):
"the rather long title already points to the structural problems of the paper and its lack of clear focus and rigid analysis"
We shortened the title to focus more on the theme and objective of the research.
"the content from lines 642-741, or parts of it, could be flagged earlier on to provide a clearer idea about the aims and objectives of the research."
We moved the in-depth analysis of the potential of places of worship and faith routes to a paragraph before the local field research. Although the research process involved the simultaneous association of evaluations related to enhancement and management and the processing of information collected from case studies, from a methodological point of view, we agree on the opportunity to identify the objectives and references of local development in the first part of the manuscript.
"The article draws on some of the existing literature on religious tourism but leaves its focus on Italian literature unexplained. Indeed, there is very little scope for discussion and critical readings of already existing studies and debates outside of Italy."
Even if the research topic is related to the Marian sphere, the broader objectives are those of investigating the local components that have determined the spread of the cult and the perspectives of valorizing that phenomenon in that specific area. For this reason, we initially limited the definition of the state of the art to experiences that are geographically or culturally similar. However, in a broader and more in-depth scientific guise, following your suggestions, we considered it useful to learn about the research in progress in other areas and to integrate the bibliography by including miscellanies and essays that refer to the international debate on Marian pilgrimage and sanctuaries dedicated to the Virgin.
"The author appears not to address a wider and cross-disciplinary readership but rather a more specialised audience. The author should place his/her analysis in the context of a particular debate and/or academic subfield. This would help to sharpen the analytical focus of the article and make it more accessible to a wider readership."
The manuscript engages with the ongoing discourse on enhancing religious itineraries and Marian pilgrimage, drawing upon a range of recent studies to delineate the methodological framework within which the local investigations are situated. The journal's scientific context and the monograph's theme inherently cater to a specialized readership. Nonetheless, in the revised manuscript, the title and the restructured text were specifically crafted to address the reviewer's suggestion.
"Editing of English language required."
Due to a final editing error, the text under review contained a paragraph in Italian. We have subjected the manuscript to a thorough review by a native English (UK) translator.
We believe that these changes have significantly improved the clarity, coherence, and overall quality of our work.
Thank you once again for your invaluable contributions. Your expertise and guidance have been instrumental in advancing our research and strengthening the manuscript.
Reviewer 3 Report
Comments and Suggestions for Authors
The article is an interesting study of material culture, devotion, and quality enhancement in cultural/religious tourism of the Marian Landscape along the Appian Way in Apulia and Basilicata.
Most of the references are Italian works. This is not a complaint, just an observation.
In the theoretical part, the author can refer more broadly to the phenomenon of pilgrimages to Marian sanctuaries and point to several examples - not only in Europe (vers. 118-119).
"Marian Studies" Vol. 51 (2000) With the Mother of the Lord: On Pilgrimage to the New Millennium.
The practical nature of the analysis and conclusions increases the value of the article.
Ho letto con grande interesse questo articolo! Buon lavoro.
Author Response
Dear Reviewer, first of all, I would like to express our sincere gratitude for your appreciation of our article.
With regard to your valuable suggestions, we found both to be highly beneficial in enlarging the field of studies. In particular, the research topic is related to the Marian sphere, but with the broader objective of investigating the local components that have determined the spread of the cult and the perspectives of valorizing that phenomenon. For this reason, we initially limited the definition of the state of the art to experiences that are geographically or culturally similar. However, in a broader and more in-depth scientific guise, following your suggestions, we considered it useful to learn about the research in progress in other areas and to integrate the bibliography by including miscellanies and essays that refer to the international debate on Marian pilgrimage and sanctuaries dedicated to the Virgin.
We hope that the new version of the manuscript will meet your observations.
Grazie!
Round 2
Reviewer 2 Report
Comments and Suggestions for Authors
It has been this reviewer’s pleasure to read the manuscript again. The changes made by the author(s) have significantly improved the structure of the article and thus made it more accessible. The aims and objectives of the article are now much clearer and offer plenty of food for thought.
Below a few suggestions how, in this reviewer’s opinion, the article could be further improved. This means only a few minor changes, such as inserting connecting words or sentences or referring to the case studies provided in the text, will suffice but make a difference. See 4 points below:
1 Title:
This reviewer takes the view that “material culture” in the title is perhaps misleading as no reference is made to the Material Culture debate. The way this reviewer reads the article, the author(s) seem more concerned with heritage sites (including landscapes) rather than material culture in a more general sense. It might be better to use heritage sites rather than material culture.
This reviewer notes that the title is still far too long and does not really point to what the article seems to be all about. In other words, the title does not translate into an attention-grabbing headline, which seems a pity. For example, the author(s) could insert “struggle” “harmony” or any of the words used in the text to highlight the issues being addressed. This reviewer can think of two possibilities, though there surely are much better ones:
Heritage Sites, Devotion and Quality Enhancement in Tourism: Promotion and Management of Ancient Marian Places of Worship along Italy’s Appian Way
The Struggle for Harmony in Quality Enhancement Tourism: Promotion and Management of Ancient Marian Places of Worship along Italy’s Appian Way
2 Abstract
As is the case with the title, the abstract does not really “sell” the article. This reviewer suggests to insert one or two sentences after the second sentence in the abstract to flag what is being discussed at lines 120; 133; 139; 182; 776; 821; 837; 858; 870. Arguably, this is what readers will be interested in and, in a sense, the contribution this article can make to ongoing debates.
Examples from the text that could be flagged in abstract and/or introduction:
“Interaction of subjects pursuing divergent interests and objectives”
“The dynamics governing relationships between worship and culture are not always simple and straightforward”.
And so on. See all the lines listed above.
3 Connecting Case studies with introduction, discussion, and conclusion
This reviewer cannot see any obvious connection being made between the part which presents the case studies and the rest of the text. Connecting sentences, phrases such as, for example, “as our case study XXXX shows” or “as discussed in relation to XXX” or “this illuminates/demonstrates/elucidates” and so on would be very helpful here, especially before and after the part presenting the case studies. There seems no apparent connection between these different parts and especially the move from case studies to discussion/conclusion seems abrupt and somehow disconnected.
Line 187: connecting sentence is needed here. This reviewer cannot see a connection between these two parts of the paper.
Overall, the paper could be further improved, if the author(s) link the case studies to the various points and arguments being made throughout the article. Not for comparative reasons, but to demonstrate how these case studies help us to understand the theoretical and policy points being made. This would further improve the article and make it more accessible. Just a few connecting words and sentences throughout the text will do.
4 Miscellaneous
Line 281: The added paragraph works well and explains to the reader why this is taken up.
Line 306: Something seems missing here.
Line 475???
Line 554 ???
